# MWIRGas-YOLO: Gas Leakage Detection Based on Mid-Wave Infrared Imaging

**DOI:** 10.3390/s24134345

**Published:** 2024-07-04

**Authors:** Shiwei Xu, Xia Wang, Qiyang Sun, Kangjun Dong

**Affiliations:** Key Laboratory of Photoelectronic Imaging Technology and System, Ministry of Education of China, Beijing Institute of Technology, Beijing 100081, China; 3220220555@bit.edu.cn (S.X.); 3220215081@bit.edu.cn (Q.S.); 3120220543@bit.edu.cn (K.D.)

**Keywords:** mid-wave infrared imaging, gas leak detection, global attention mechanism, small target detection layer

## Abstract

The integration of visual algorithms with infrared imaging technology has become an effective tool for industrial gas leak detection. However, existing research has mostly focused on simple scenarios where a gas plume is clearly visible, with limited studies on detecting gas in complex scenes where target contours are blurred and contrast is low. This paper uses a cooled mid-wave infrared (MWIR) system to provide high sensitivity and fast response imaging and proposes the MWIRGas-YOLO network for detecting gas leaks in mid-wave infrared imaging. This network effectively detects low-contrast gas leakage and segments the gas plume within the scene. In MWIRGas-YOLO, it utilizes the global attention mechanism (GAM) to fully focus on gas plume targets during feature fusion, adds a small target detection layer to enhance information on small-sized targets, and employs transfer learning of similar features from visible light smoke to provide the model with prior knowledge of infrared gas features. Using a cooled mid-wave infrared imager to collect gas leak images, the experimental results show that the proposed algorithm significantly improves the performance over the original model. The segment mean average precision reached 96.1% (mAP50) and 47.6% (mAP50:95), respectively, outperforming the other mainstream algorithms. This can provide an effective reference for research on infrared imaging for gas leak detection.

## 1. Introduction

Industrial gas is widely used in modern society in fields such as steel production, commercial activities, transportation, power generation, and chemical industry [1], making it a crucial component of the energy system and a significant driver of high-quality socio-economic development. However, gas leaks frequently occur during storage, use, and transportation, which can easily lead to fires and explosions [2], causing substantial loss of life and property. Therefore, achieving rapid and effective gas leak detection, along with accurate localization of the leak source and the affected area, are of paramount importance.

Traditional contact-based detection methods, such as acoustic [3], electrochemical [4], and semiconductor techniques [5], can accurately pinpoint leak locations and have low production costs. However, they are significantly affected by environmental interference signals and temperature variations [6,7]. Infrared imaging technology, as a typical non-contact detection method, offers a wide detection range and intuitive visualization, enabling remote dynamic monitoring [8,9]. Considering that industrial gas predominantly exhibits characteristic absorption spectra in the mid-wave and long-wave infrared bands, post-processing with image analysis algorithms allows for the identification of gas leak locations and diffusion trends, thereby enhancing the detection efficiency and saving substantial time and labor. This technology is widely used for industrial gas leak detection. However, the non-solid nature and variable characteristics of gas plumes [10] indicate that the visual effects of gas infrared images are highly influenced by environmental changes. This often results in a gas plume with blurred contours and low contrast against the background in infrared images, making it difficult to observe effectively.

Infrared imaging systems are categorized into active [11] and passive [12] systems based on the radiation source. Unlike active imaging systems, which require additional light sources to provide radiation, passive infrared imaging systems rely solely on ambient background radiation within the field of view. This makes them smaller, more cost-effective, and easier to implement in portable automated detection. Consequently, imaging technologies based on infrared focal plane arrays are more widely used [13]. As the core component of infrared imaging systems, infrared detectors are classified into thermal and photon detectors based on their energy conversion methods. Additionally, they are divided into cooled and uncooled detectors depending on their operating temperature and cooling requirements. Cooled mid-wave infrared detectors [14], which detect infrared radiation through the photoelectric effect generated by the interaction between incoming photons and sensitive materials, offer rapid response and high reliability. These detectors provide high sensitivity and fast response imaging, detecting minute temperature differences and radiation disparities, making them particularly advantageous for gas leak detection tasks.

With the advancement of deep learning, artificial intelligence has been widely applied in the field of computer vision, demonstrating strong vitality in areas such as autonomous driving [15], medical image processing [16], and equipment inspection [17]. Owing to their effective feature extraction capabilities, deep learning-based methods have become a significant direction in gas leak detection research, with models suitable for continuous gas detection. However, several challenges remain: (a) For computer vision tasks, large-scale, high-quality, and well-annotated datasets are crucial for model training, yet there is currently no unified dataset for comparing the performance of different algorithms. (b) Infrared image quality, affected by infrared detectors, significantly impacts the detection work. A low-concentration gas plume is easily confused with image noise, which reduces the detection efficiency. (c) A gas plume in infrared images often has blurred edges, low background contrast, and poor discernibility, resulting in limited usable spatial features and challenging gas feature extraction.

The You Only Look Once (YOLO) series of algorithms, a convolutional neural network-based object detection framework, has demonstrated excellent performance in terms of generalization and real-time capabilities for computer vision tasks. It is widely used in object detection research and can be considered a viable solution for gas leak detection using infrared imaging.

The main contents of this paper are as follows:A high-sensitivity and high-response imaging effect was achieved using a cooled mid-wave infrared (MWIR) imager. A dataset labeled with gas leak segmentation, MWIRGas-Seg, was collected and underwent visual classification and small target counting.For the task of gas leak detection in MWIR imaging, an algorithm based on YOLOv8-seg is proposed. This algorithm, named MWIRGas-YOLO, effectively detects and segments gas leaks within a given scene.A global attention mechanism was introduced during the feature fusion stage to reduce image information dispersion, enhance gas plume localization, and improve the extraction of small target gas plume features. Transfer learning was applied using a visible light smoke dataset with similar characteristics to make the pre-trained model more adept at handling and extracting gas features.Experimental validation confirms that MWIRGas-YOLO achieves more effective feature extraction and fusion of infrared gas plume targets, outperforming the original YOLOv8-seg and several typical image detection and segmentation algorithms. It is suitable for infrared gas image detection tasks.

## 2. Related Works

### 2.1. Gas Leakage Detection

Early gas leak detection algorithms treated the gas plume as a whole, utilizing traditional moving object detection methods such as frame differencing [18], optical flow [19], and background modeling [20,21]. For instance, Lu et al. [22] detected gas leak areas using a mixture of Gaussian models, and Shen et al. [23] investigated the application of three optical flow estimation algorithms to measure fluid flow velocity. Weng et al. [24] proposed a gas plume detection algorithm based on a scale-invariant feature transform and a support vector machine for image sequences filtered by frame differencing. Si et al. [25] developed a method for detecting sulfur hexafluoride ($SF_{6}$) gas by combining gas absorption peaks with binocular filter differentiation to eliminate noise interference. Yan et al. [26] proposed a natural gas pipeline leakage detection method that converts data from multiple sensors into time–frequency images and employs multi-source, multi-modal feature fusion. However, due to the inherently high noise and dynamic background conditions of infrared images, traditional motion object detection methods struggle to adapt to practical gas leak detection tasks, limiting their real-world applicability.

In recent years, with the successful application of deep learning methods in video image processing and object detection, researchers have begun to explore deep learning algorithms for infrared imaging gas leak detection. In 2019, Wang et al. [27] applied CNNs to detect methane leaks, using a FLIR GF320 thermal camera to collect and label the first large-scale methane leak video dataset, GasVid. This dataset covers various leak sources and leak sizes, allowing for the testing of the detection efficiencies of different CNN architectures. The study also compared the fixed background subtraction, moving average background subtraction, and Gaussian mixture background methods. Two years later, the same team developed VideoGasNet [28] for gas leak detection in videos, extending the 2D CNN input of 2D image data to a 3D CNN by incorporating the temporal dimension. Their results demonstrated that the 3D CNN (VideoGasNet) had improved robustness, with leak/no-leak detection accuracy near 100%, and classification accuracy for large, medium, and small leak sizes reaching 78.2%. However, while the study achieved high accuracy in detecting the presence of gas leaks within a scene, it did not specify the exact location or quantify the leaks. Additionally, the dataset was collected with a single clear-sky background and no other scene disturbances, raising concerns about its generalizability.

Shi et al. [29] integrated the two-stage detection model Faster R-CNN with optical gas imaging (OGI) technology, utilizing model transfer methods to achieve real-time automatic hydrocarbon leak detection. Bin et al. [30] proposed a gas detection framework that integrates foreground information based on background subtraction methods and deep neural networks. Park et al. [31] employed U-Net as a fully convolutional network to accomplish semantic segmentation of infrared gas images. Wang et al. [32] used a Plug617 uncooled infrared detector to image SF6 gas and proposed a gas semantic segmentation method based on the DeeplabV3+ model. Bhatt et al. [33] introduced a spatio-temporal U-Net architecture to perform pixel-level gas mask extraction from input image frame sequences. Lin et al. [34] proposed the semantic segmentation model 2.5D-Unet, which enhances the network’s capability to represent the appearance and motion of leaking gas. Gu et al. [35] demonstrated that a two-dimensional Gaussian distribution could better cluster gas plume pixels, and that training on synthetic datasets can be effectively applied to real-world scenarios. Badawi et al. [36] demonstrated that utilizing raw data with deep learning methods allows algorithms to learn and discern features autonomously.

Significant progress has been made in infrared imaging for gas leak detection. However, current detection algorithms primarily focus on single scenarios with a clearly visible gas plume without accounting for real-world conditions where targets have blurred contours and low contrast. Moreover, there is limited research on the use of cooled infrared detectors for gas leak detection. Additionally, the application of classic image segmentation algorithms such as YOLACT [37], SOLO [38], SOLOv2 [39], QueryInst [40], and YOLO to infrared gas images is sparse. Few comparative analyses of various existing algorithms have been conducted on a common dataset.

### 2.2. YOLO Detection Models

YOLO, as a typical representative of single-stage detection models, is widely applied in real-time object detection. Redmon et al. first proposed YOLOv1 [41] in 2016 to address the high computational complexity of two-stage models, treating object detection as a regression problem where the entire image serves as a network input. A neural network predicts bounding box positions and class labels. Subsequent advancements including YOLOv2 [42] and YOLOv3 [43] incorporated superior backbone networks and feature pyramids, significantly enhancing detection performance.

In 2020, YOLOv4 [44] and YOLOv5 introduced more advanced detection strategies such as data augmentation and replacing the PAN structure with FPN-PAN, improving detection efficiency while maintaining real-time capability. Continuing advancements led to YOLOX [45], YOLOv6 [46], and YOLOv7 [47], progressively enhancing detection performance in general object detection tasks. In 2023, researchers introduced YOLOv8, which not only improved on structure and loss functions but also integrated instance segmentation capabilities inspired by YOLACT. This enhancement delineates object contours in images, thereby diversifying the tasks for object detection. Consequently, applying the YOLOv8 model to gas detection and segmentation tasks is feasible.

## 3. Datasets

### 3.1. Data Acquisition

In this paper, a cooled mid-wave infrared imaging system was employed to investigate gas leak detection using the image acquisition process illustrated in Figure 1. The detection principle of infrared imaging for gas leakage involves two main components. The first component is the infrared spectrum of the background radiation/reflection that reaches the front boundary of the gas plume and detector. The second component comprises background infrared radiation, partially absorbed by the gas plume, and the radiation emitted by the gas plume itself, which, together, reach the detector through the atmosphere. The presence of the gas plume causes differential radiation, after gas absorption, to exceed the sensitivity threshold of the detector, resulting in a detectable radiative difference in the relevant infrared absorption bands. This differential radiation is focused onto the infrared detector through optical lenses, producing a grayscale image that represents the radiative contrast of the target infrared scene.

The specific parameters of the cooled mid-wave infrared imaging system are listed in Table 1. Methane and HFC gas were employed in these experiments. Methane, a typical industrial gas, exhibits a significant absorption peak at 3.3 μm [48], which falls within the MWIR range and matches the detection band of the detector used in the experiment. HFC gas is a portable refrigerant that is colorless, odorless, and highly safe. During its release, it undergoes noticeable vaporization and heat absorption, resulting in a radiative contrast within the infrared absorption bands.

The study of image segmentation requires a large number of data samples to train deep learning networks. However, there is currently no open-source dataset for infrared gas image segmentation. Therefore, to apply image segmentation to gas leak detection, this paper established an MWIRGas-Seg dataset for mid-wave infrared gas leak segmentation. This dataset includes multiple indoor and outdoor gas leak scenes with various leak sources, while also recording weather conditions, environmental temperatures, wind speeds, and detection distances.

The collected images encompassed various detection distances (7, 10, 20, 30, 33, 40, and 50 m), leak rates (0.5, 1, 2, 5, and >5 L/min), and 10 experimental scenarios, totaling 35 sets of image sequences. Representative frames of different leak categories are displayed in Figure 2. X-AnyLabeling 2.3.0 [49] was used to annotate 7187 sets of segmentation data based on pixel-level differences between the gas leak areas and background areas.

### 3.2. Dataset Statistics

Owing to the influence of the gas properties and environmental factors, the grayscale differences observed in the images do not directly correlate with the actual leak distance and volume. Therefore, this paper categorized infrared gas images into three classes—dense, thin, and indiscernible. Figure 3 illustrates the typical representatives of each class.

Chen et al. [50] defined small objects as those with a bounding box area to image area ratio between 0.08% and 0.58%. The pixel information of the mask in our dataset was statistically analyzed, as shown in Figure 4, to calculate the number of small and normal objects in the dataset, as presented in Table 2. The dataset comprised 62.28% normal objects and 37.72% small objects.

## 4. Methods

### 4.1. YOLOv8-Seg Model

The YOLO series of methods, as one-stage object detection algorithms, utilize a backbone network to extract features, fuse multi-scale features, and then output target detection boxes through multiple detection heads. This approach achieves an excellent detection accuracy and real-time performance. The YOLOv8 model integrates methods from YOLACT, adding a segmentation branch on top of the existing detection branches to combine detection boxes and segmentation results, thereby achieving instance segmentation of targets. The backbone enhances the feature extraction capabilities using a lightweight C2f module and incorporates an SPPF layer at the end to extract features with different receptive fields, making the network more suited for targets of different scales. The neck section employs a dual-stream Feature Pyramid Network (FPN) structure that aggregates feature information from top to bottom, fusing high- and low-level feature information via upsampling to compute prediction feature maps. A Path Aggregation Network (PAN) introduces lateral connections to enhance semantic feature information and fusion capabilities. The head section adopts an anchor-free decoupling head, which inputs position and category information from the feature map into the detection and classification branches to compute position and category, respectively. Predictions for small-, medium-, and large-scale targets were generated based on the fused P3, P4, and P5 feature maps.

This paper proposes the MWIRGas-YOLO model for mid-wave infrared imaging gas leak detection tasks, which incorporates a global attention mechanism and a small object detection layer based on the YOLOv8-seg model. The network structure is illustrated in Figure 5.

### 4.2. Global Attention Mechanism

In the context of infrared imaging gas leak detection, weak-concentration gas plume targets are often confused with background noise, posing challenges for detection and segmentation. To address this issue, adding the global attention mechanism can assign different weights to different parts of the input feature map, allowing the network to focus fully on gas plume targets while ignoring irrelevant background information, thus improving the detection accuracy.

The global attention mechanism (GAM) [51] consists of the channel attention submodule $M_{c}$ and spatial attention submodule $M_{s}$. The channel attention submodule preserves information in three different dimensions and utilizes a two-layer Multilayer Perceptron (MLP) to amplify spatial information interaction across dimensions, thereby enhancing feature representation capability (as shown in Figure 6). The spatial attention submodule focuses on spatial information, using two convolutional layers for spatial information fusion and fully learning spatial features (as shown in Figure 7).

The entire process is illustrated in Figure 8. For a given input feature map $F_{1}$, after channel attention, the resulting feature is denoted as $F_{2}$ and the final output feature is denoted as $F_{3}$; the intermediate state $F_{2}$ and output $F_{3}$ are defined as follows:(1)$$F_{2}=M_{c}\left(F_{1}\right)⊗F_{1}$$
(2)$$F_{3}=M_{s}\left(F_{2}\right)⊗F_{2}$$
where ⊗ indicates that multiplication is performed element by element.

In the YOLOv8 model, the backbone is responsible for extracting feature information from images, whereas the neck section focuses on better utilizing the extracted features for feature fusion. For gas leak detection tasks, noise in infrared images can significantly affect the feature extraction of gas plume targets. Adding an attention mechanism at the backbone feature extraction stage amplified the noise impact while reinforcing the gas plume target features. Therefore, this paper adds a global attention mechanism at the neck feature fusion stage to enhance the feature fusion capability of gas plume targets and improve the detection accuracy.

### 4.3. Small Target Detection Layer

The original YOLOv8 model was equipped with three detection heads, enabling multi-scale object detection. The detection sizes were P3/8, P4/16, and P5/32, which correspond to feature map sizes of 80 × 80, 40 × 40, and 20 × 20, respectively. These feature maps are responsible for detecting objects of sizes 8 × 8, 16 × 16, 32 × 32, and larger However, due to the presence of small targets within gas plume clusters, which are often influenced by environmental interference, deeper feature maps struggled to capture the features of small objects effectively. Consequently, the original model exhibited a poor performance in detecting small targets.

This paper proposes the addition of a small object detection layer to the original network, specifically a P2/4 detection layer with a size of 160 × 160. This modification enhances the semantic information and feature representation capabilities of small objects by incorporating supplementary fusion feature layers and an additional detection step. The 80 × 80 scale layer from the fifth layer (P2) in the backbone was stacked with an upsampled feature layer in the neck. After processing with C2f and upsampling, this results in a deep semantic feature layer containing small object feature information. This layer is then further stacked with the shallow positional feature layer from the third layer in the backbone, creating a comprehensive 160 × 160 scale fusion feature layer that improves the expression of semantic and positional information for small objects. Finally, this enriched feature layer is sent through a C2f module to an additional decoupled head.

The enhancement in the head section allows the small object feature information to continue propagating along the downsampling path to the other three scale feature layers. This strengthens the feature fusion capability of the network, and the introduction of an additional decoupled head expands the detection range for gas plume targets. These improvements in the detection accuracy and range enable the network to more precisely identify small gas plume targets within the scene.

### 4.4. Transfer Learning

The objective of transfer learning is to apply knowledge or patterns learned from one domain or task to a different but related domain or problem [52]. Given that gas plume targets and smoke share similar uncertain contour characteristics, this paper employed a smoke dataset with similar features for transfer learning pre-training. This approach yields a prior model capable of effectively extracting features from gas plume targets. The smoke images used for this purpose were sourced from publicly available online datasets.

Testing the model, which was trained on a visible light smoke image dataset, directly on infrared gas images revealed that the smoke model could effectively identify some gas regions, as shown in Figure 9. This demonstrates that infrared gas plumes and smoke share similar features and confirms that the prior model has the capability to learn and extract features for gas plume targets.

## 5. Results and Analysis

### 5.1. Experimental Configuration and Model Setup

The experiment was conducted on an Ubuntu 20.04.6 LTS operating system using a GeForce RTX 3090 GPU, Python 3.8, and CUDA 12.1. The deep learning framework employed was PyTorch 2.2.2. The training parameters are presented in Table 3. The dataset was divided into a training set and a validation set in an 8:2 ratio, with a training set comprising 5749 images and a validation set comprising 1438 images.

### 5.2. Evaluation Metrics

To evaluate the performance of the algorithm in infrared imaging gas leak detection and objectively compare the detection effectiveness of different segmentation algorithms, this experiment employed precision, recall, and mean average precision (mAP) as performance metrics.

Precision refers to the probability of correctly predicting positive samples among the predicted positive samples:(3)$$P=\frac{TP}{TP+FP}$$
where TP and FP represent the number of true and false positive samples in the predicted positive samples, respectively.

Recall is the probability of correctly predicting a positive sample from all predicted samples:(4)$$R=\frac{TP}{TP+FN}$$
where FN is the number of false negative samples predicted as negative.

mAP is the average accuracy of all the detection classes:(5)$$mAP=\frac{\sum_{0}^{N}AP_{n}}{N}$$

*N* represents the total number of classes and $AP_{n}$ is the average accuracy rate of class *n*, which is obtained by calculating the area under the precision–recall rate curve. The mean value is denoted as mAP@0.5, indicating the average accuracy rate when the threshold of the IoU parameter is set to 0.5, and mAP@0.5:0.95 indicates the average accuracy rate at different IoU thresholds (from 0.5 to 0.95, step size 0.05).

### 5.3. Experimental Analysis of Global Attention Mechanism

To validate the effectiveness of incorporating the GAM in the neck section for the feature fusion of gas plume targets, a comparative experiment was conducted by embedding the GAM at different positions in the YOLOv8-seg model’s backbone and neck. The experimental results are listed in Table 4.

The table shows that embedding GAM in the backbone feature extraction part enhances gas plume feature extraction but also increases the interference impact on feature extraction, thereby decreasing the detection accuracy and rate. Feature extraction before and after embedding the GAM is illustrated in Figure 10a. Introducing the GAM in the neck section focuses more on gas plume feature fusion, resulting in an improved algorithm accuracy and mAP. Figure 10b compares the feature maps before and after embedding the GAM in the neck section.

### 5.4. Experimental Analysis of Small Target Detection Layer

Infrared imaging gas plumes often exhibit uneven concentrations, owing to the influence of environmental wind speeds, making it challenging to visually form a cohesive whole. The addition of a fourfold downsampling detection layer aims to enhance the ability of the model to effectively detect weakly concentrated small gas plumes. Figure 11 illustrates the detection comparison before and after the addition of the small object detection layer. It can be observed that the addition of fourfold downsampling corresponding to the small object detection layer effectively mitigates issues of missed detection caused by small gas plume sizes, thereby improving detection accuracy.

### 5.5. Experimental Analysis of Transfer Learning

The diffusion of infrared imaging gas plumes and visible light smoke adheres to the principles of fluid mechanics, with similar contour characteristics. This paper leveraged feature learning from visible light smoke images to obtain a prior model capable of learning gas plume features. Figure 12 presents the visualization results of training the infrared gas plume model based on the original YOLOv8-seg model and smoke model as the base models. Training the infrared gas plume model using the smoke model demonstrates a clear prior, leading to a noticeable improvement in model accuracy and precision.

### 5.6. Ablation Experiment

To further validate the differences in enhancing the model segmentation performance through the GAM, small object detection layer, and smoke image transfer learning guidance, ablation experiments were conducted on the same dataset. The experiments aimed to verify the impact of embedding the three modules on model segmentation accuracy. Six sets of ablation experiments involved single modules or combinations of two modules, with the final set comprising the complete algorithm proposed in this paper. Table 5 presents the results. The experimental results demonstrate that both the GAM attention mechanism and the small object detection layer significantly improve the precision. Additionally, the use of smoke image transfer learning guidance significantly enhances segmentation accuracy.

### 5.7. Contrast Experiment

To validate the superiority and effectiveness of the proposed algorithm, we selected representative image segmentation algorithms including Mask R-CNN, YOLACT, SOLO, SOLOv2, and QueryInst for comparative experiments with our model. We calculated the target detection and image segmentation of mAP50 and mAP50:95 for each model. To ensure the validity of the results, all the experiments were conducted on a previously described experimental platform using the same training and validation datasets.

As shown in Table 6, compared with the other six models, the proposed model achieved the highest target detection mAP50 and mAP50:95, reaching 97.3% and 59.9%, respectively. In addition, the proposed model achieved the highest image segmentation of mAP50 and mAP50:95, achieving 96.1% and 47.6%, respectively.

To visually compare the robustness of our model in segmenting infrared gas images, we present the segmentation results of the original model, the five comparison models, and the improved model in Figure 13. The first and second rows display the original images and their corresponding ground truth segmentations. Rows three–nine show the visualization results of the various models. From the visualizations, it is evident that in columns two and five, the YOLOv8-seg model incorrectly segmented a single gas plume into two separate plumes, whereas the MWIRGas-YOLO model accurately segmented them. In the image in column four, due to the low contrast between the gas plume and background, the YOLOv8-seg model failed to detect the gas leak effectively. In contrast, the MWIRGas-YOLO model, with the addition of a small target layer and global attention mechanism, more effectively extracted and fused gas features, enabling the detection of faint gas leak. A detailed examination of columns two and five reveals that our method segments the contours of morphologically variable gas plume more accurately, producing results closer to the ground truth. In summary, our designed model demonstrated superior detection and segmentation performance, making it more suitable for detecting and segmenting infrared gas images.

## 6. Conclusions

In this paper, a cooled mid-wave infrared thermal imager was used to create the MWIRGas-Seg dataset for mid-wave infrared gas leak segmentation. The analysis was conducted based on experimental scenarios and visual judgments. A model specifically designed for mid-wave infrared gas leak detection and segmentation, MWIRGas-YOLO, was proposed to achieve effective detection of leaking gas and segmentation of gas plumes in the scene. Compared with typical image segmentation models, the proposed model demonstrated superior performance in gas target detection and segmentation. Moreover, our method focused directly on studying gas plume features in infrared images, embedding the GAM to enhance the fusion of features, thereby focusing on gas cloud targets. Supplementary small target detection layers effectively address issues of missed small-sized gas cloud targets. The model is pre-trained using smoke images, showing significant prior knowledge in learning infrared gas cloud characteristics. The experimental results indicate that the MWIRGas-YOLO model designed in this paper exhibits superior detection and segmentation performance, making it suitable for infrared gas image detection and segmentation tasks. Importantly, our proposed method is versatile and does not depend on specific infrared detectors; thus, it can be applied broadly in the field of infrared imaging for gas leak detection.

Future studies will continue to explore this technology by expanding the dataset to represent a broader range of real-world leak scenarios. The goal was to make the algorithm more applicable for leak detection in real-world scenarios. Additionally, as gas leak processes are continuous, future efforts will investigate incorporating temporal information from videos and exploring sequence modeling to address gas leak issues, further optimizing the algorithm for enhanced accuracy.

## Figures and Tables

**Figure 1 sensors-24-04345-f001:**
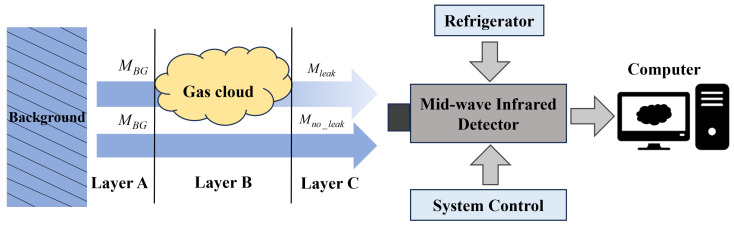
Three-layer radiation transfer model and gas infrared imaging detection system. $M_{BG}$ is the background spectral radiation, $M_{leak}$ and $M_{no\_leak}$ are gas path and non-gas path radiation.

**Figure 2 sensors-24-04345-f002:**
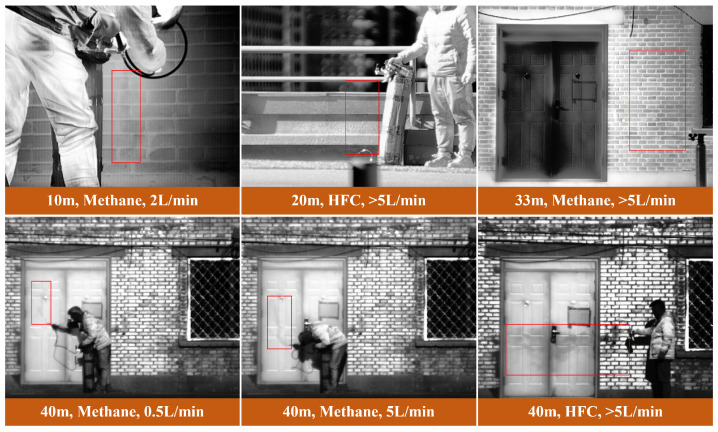
Infrared gas image samples.The red bounding box represents the leaked gas. Below the image are the leak distance, gas type, and leak rate.

**Figure 3 sensors-24-04345-f003:**
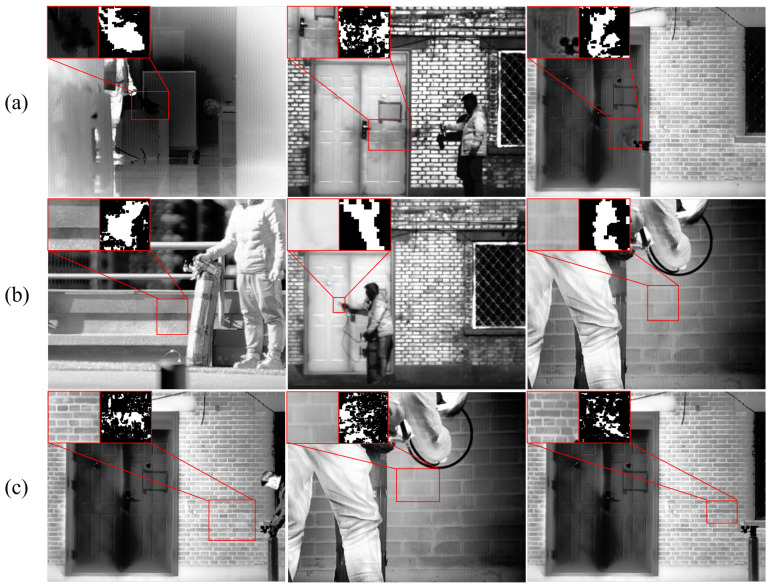
Infrared gas images categorized into three types; the top left of the image shows enlarged views of the gas region image and its corresponding frame difference image. (**a**) Dense, where the gas plume in the scene is easily observable at a glance. (**b**) Thin, requiring careful observation of local areas to detect the gas plume. (**c**) Indiscernible, where the gas plume cannot be directly observed in a single frame and requires observation of multiple frames before and after to identify its position.

**Figure 4 sensors-24-04345-f004:**
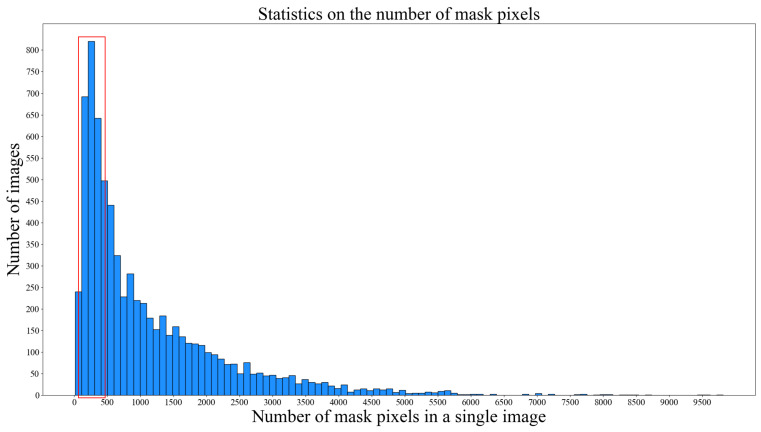
Histogram of mask pixel count. A mask pixel count ranging from 65 to 475 in a single image is considered a small object.

**Figure 5 sensors-24-04345-f005:**
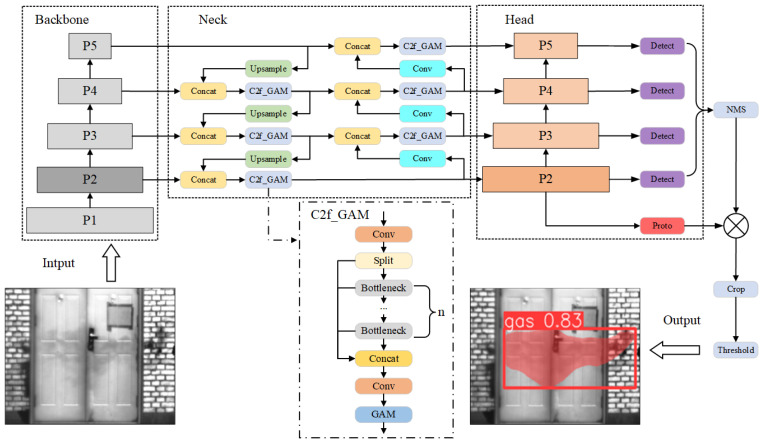
MWIRGas-YOLO network architecture.

**Figure 6 sensors-24-04345-f006:**
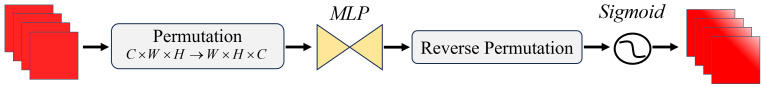
Channel attention submodule.

**Figure 7 sensors-24-04345-f007:**
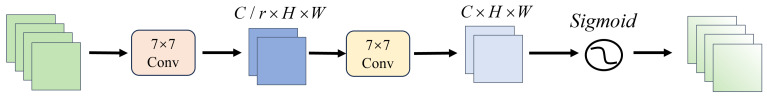
Spatial attention submodule.

**Figure 8 sensors-24-04345-f008:**
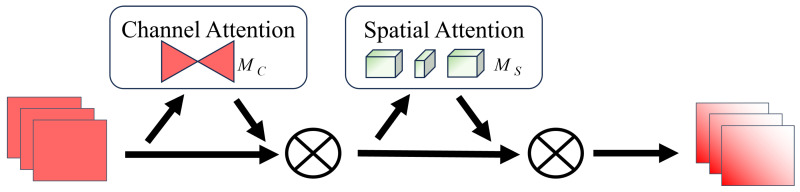
Global attention mechanism.

**Figure 9 sensors-24-04345-f009:**
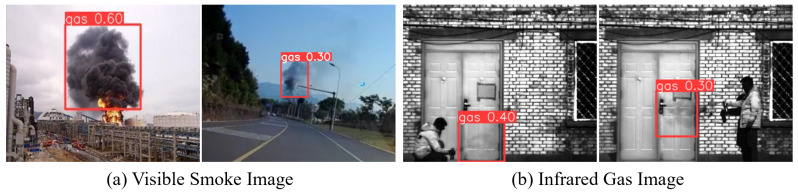
Prior model detection results. (**a**) Visible smoke image detection results. (**b**) Infrared gas image detection results.

**Figure 10 sensors-24-04345-f010:**
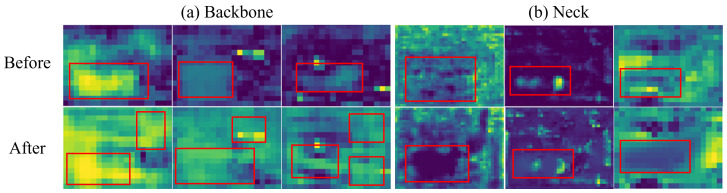
The feature map before and after embedding GAM, and the red bounding box is the activation of the target feature. (**a**) Results of backbone comparison. (**b**) Results of neck comparison.

**Figure 11 sensors-24-04345-f011:**
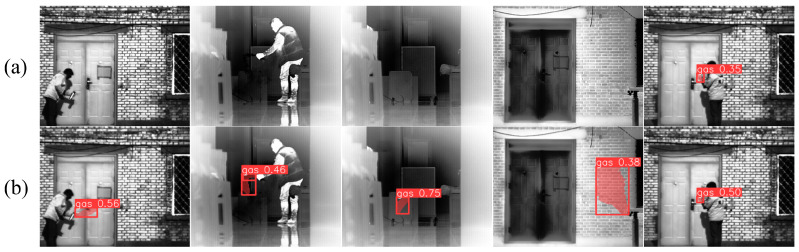
Test comparison diagram. (**a**) YOLOv8-seg test result. (**b**) MWIRGas-YOLO test result.

**Figure 12 sensors-24-04345-f012:**
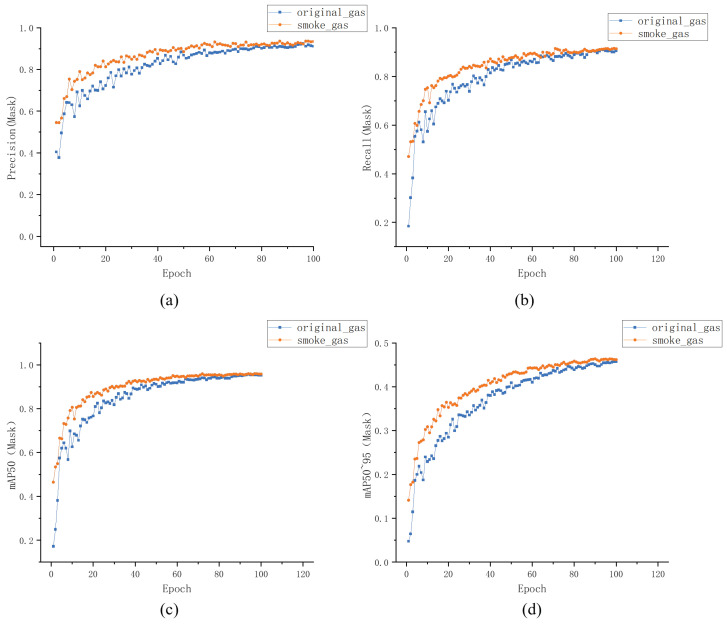
Visual comparison of training process. (**a**) Precision curve. (**b**) Recall curve. (**c**) mAP50 curve. (**d**) mAP50:95 curve.

**Figure 13 sensors-24-04345-f013:**
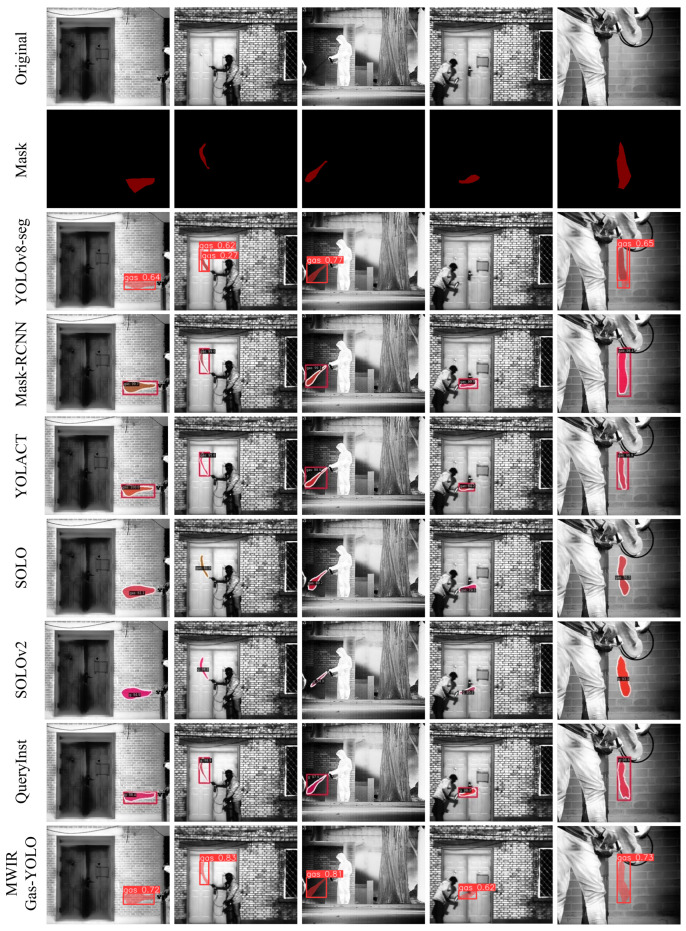
Visualization of detection and segmentation results by different methods on MWIRGas-Seg dataset.

**Table 1 sensors-24-04345-t001:** Main performance parameters of infrared imaging device.

Performance Metrics	Parameters
Array format	320 × 256
Infrared detector type	Cooled infrared detector
Image size	30 μm
NETD	<20 mK
Spectral range	3.2–3.4 μm
Focal length	100 mm
Frame per second	30 fps

**Table 2 sensors-24-04345-t002:** Statistics of dataset categories.

Object Statistics	Number of Objects
Small object	2711
Normal object	4476
Total objects	7187

**Table 3 sensors-24-04345-t003:** Model training hyperparameter settings.

Hyperparameter Options	Parameters
Imgsz	640
Batch_size	16
Epochs	100
Optimizer	SGD
Initial Learning Rate	0.01
Learning Rate Float	0.01
Momentum	0.937

**Table 4 sensors-24-04345-t004:** Comparison results. The best result is indicated in bold.

Optimized Method	Precision	mAP50	mAP50:95
YOLOv8-seg	91.4	95.3	45.7
+GAM_All	90.8	94.2	44.5
+GAM_Backbone	90.4	94.7	44.6
+GAM_Neck	**91.5**	**95.4**	**46.1**

**Table 5 sensors-24-04345-t005:** Ablation experiment results. The best result is indicated in bold.

Optimized Method	GAM	Small Target Detection Layer	Transfer Learning	Precision	mAP50	mAP50:95
YOLOv8-seg				91.4	95.3	45.7
YOLOv8-seg-I	✓			91.5	95.4	46.1
YOLOv8-seg-II		✓		92.7	95.8	46.2
YOLOv8-seg-III			✓	93.5	96.0	46.2
YOLOv8-seg-IV	✓	✓		91.8	95.3	46.6
YOLOv8-seg-V	✓		✓	92.2	96.0	46.4
YOLOv8-seg-VI		✓	✓	**93.6**	95.1	45.4
YOLOv8-seg-VII	✓	✓	✓	92.3	**96.1**	**47.6**

**Table 6 sensors-24-04345-t006:** Contrast experiment results. The best result is indicated in bold.

Model	mAP50_bbox	mAP50:95_bbox	mAP50_mask	mAP50:95_mask
YOLOv8-seg	96.7	58.2	95.3	45.7
Mask R-CNN	89.4	46.2	84.7	34.1
YOLACT	94.8	53.0	92.0	39.2
SOLO	-	-	86.4	33.3
SOLOv2	-	-	90.5	37.6
QueryInst	90.4	47.1	83.8	33.0
MWIRGas-YOLO	**97.3**	**59.9**	**96.1**	**47.6**

## Data Availability

Data are contained within the article.

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
