# Peer review of "MWIRGas-YOLO: Gas Leakage Detection Based on Mid-Wave Infrared Imaging"

_sensors, 2024, doi:10.3390/s24134345_

Round 1

Reviewer 1 Report

Comments and Suggestions for Authors

The authors present the gas leakage detection method using MIR imaging with the modified YOLO algorithm. It is interesting to apply the technique with the practical images with gas plume targets. This manuscript needs to make several revisions before considered for acceptance.

1 The novelty and impact should be further clarified. In the abstract and introduction sections, it is not clear for readers about the differences and benefits of this MWIRGas-YOLO compared to the conventional YOLO or YOLOv8-Seg.

2 The structure of the manuscript should be improved. It is strange to have the separated Section 2 Related Works. This content is normally included in the introduction section.

3 The authors should focus on the discussion of YOLO and elaborate the literature review in the last paragraph of Section 2 Related Works, which could help with the illustration of novelty and impact.

4 Once the MIR detector is changed, will the performance of the MWIRGas-YOLO be degraded? What about the universality of this algorithm?

5 It is difficult to observe the information in Fig13.

Comments on the Quality of English Language

 Moderate editing of English language required

Reviewer 2 Report

Comments and Suggestions for Authors

In the article "MWIRGas-YOLO:Gas Leakage Detection Based on Mid-wave Infrared Imaging" the authors proposed the new method for Gas Leakage Detection based on the using of cooled detectors and modified algorithms of neural network learning. Additionally the database of infrared and visible images were collected and used.

In the abstract the author wrote "Using a cooled mid-wave infrared imager to collect gas leak images, the experimental results show that the proposed algorithm significantly improves the performance over the original model", however the influence of the detector on the results is not discussed.

The authors declared that improvements of YOLOv8-seg algorithm results in better accuracy. Particular they present results in Tables 4 and 5. However, increasing of different estimated values is so small that it appears to be in margin of errors. Such margins can be estimated if the neural network is trained on different data sets (data for training and testing will be selected in, say, 10 different ways) and the dispersion of values of precisions and average accuracy would be calculated. Of course, it has a sense if the process of training is not very long.

At whole, the article can be published in Sensors after taking into account the mentioned comments.

Some minor issues.

Line 11. "… with prior knowledge of infrared gas". A noun in the last place seems to be missed.

Line 216 "… of thr channel attention submodule…" Probably should be "… of THE channel…"
